# Topological classification of cycloadditions occurring on-surface and in the solid-state
Juan Li[1,2,12], Amir Mirzanejad[3,12], Wen-Han Dong ●[4,12], Kun Liu[5], Marcus Richter ●[5], Xiao-Ye Wang ●[6,7], Reinhard Berger[5,8], Shixuan Du ●[4], Willi Auwärter ●[9], Johannes V. Barth ●[9], Ji Ma ●[5], Klaus Müllen[6], Xinliang Feng ●[5], Jia-Tao Sun ●[10] ✉, Lukas Muechler[3] ✉ & Carlos-Andres Palma ●[4,11] ✉

The study of cycloaddition mechanisms is central to the fabrication of extended $sp^2$ carbon nanostructures such as graphene nanoribbons and spin chains. Reaction modeling in this context has focused mostly on putative, energetically preferred, exothermic products with limited consideration for symmetry allowed or forbidden mechanistic effects. To classify and optimize allowed reaction mechanisms, modern topological tools can be explored. Here, we introduce a scheme for classifying symmetry-forbidden reaction coordinates in Woodward-Hoffmann correlation diagrams. We show that topological classifiers grant access to the study of reaction pathways and correlation diagrams in the same footing, for the purpose of elucidating mechanisms and products of polycyclic aromatic azomethine ylide (PAMY) cycloadditions of pentacene-yielding polycyclic aromatic hydrocarbons with an isoindole core in the solid-state and on surfaces, as characterized by mass spectrometry and scanning tunneling microscopy, respectively. By means of a tight-binding reaction model and broken-symmetry density functional theory (DFT), we find topologically-allowed pathways for an endothermic reaction mechanism. Our work unveils topological classification as a crucial element of reaction modeling for nanographene engineering, and highlights its fundamental role in the design of cycloadditions in on-surface and solid-state chemical reactions, while underscoring that exothermic pathways can be topologically-forbidden.

Cycloaddition reactions[1,2] are cornerstones in carbon nanomaterial engineering. Early examples include Diels-Alder[3,4] $[4+2]$ and Prato-type[5] $[3+2]$ cycloadditions in solution environments, and Huisgen-type[6] $[3+2]$ as well as related $[2+2]$ Bergman[7,8] cyclization on surfaces. With the advent of nanographene synthesis[9–12], a new chapter in organic chemistry has opened up, seeking a modular, highly selective, and high-yield reactions at interfaces without byproducts, in particular cycloadditions[13–39] for the fabrication of extended conjugated macromolecules[40]. This endeavor

embodying click-chemistry[41] has notably driven the adaptation of a large variety of organic reactions at interfaces, including Ullmann coupling, Glaser coupling and polycondensations. Recently, we have shown that polycyclic aromatic azomethine ylides[42–48] (PAMYs, Fig. 1a) can be employed to form diaza-hexabenzocoronenes and N-containing polycyclic aromatic chains[25,49–51] in the solid-state and on surfaces, opening an avenue to cycloaddition polymerizations for extended polycyclic aromatic hydrocarbons (PAHs) or related nanographenes. In solution, the PAMY

[1]School of Interdisciplinary Science, Beijing Institute of Technology, 100081 Beijing, China. [2]Energy and Transportation Domain, Beijing Institute of Technology, 519088 Zhuhai, China. [3]Department of Chemistry, Penn State University, 108 Chemistry Building, 16802 University Park, USA. [4]Beijing National Laboratory for Condensed Matter Physics and Institute of Physics, Chinese Academy of Sciences, 100190 Beijing, China. [5]Chair for Molecular Functional Materials, Center for Advancing Electronics Dresden (cfaed), Faculty of Chemistry and Food Chemistry, Technische Universität Dresden, Mommsenstr. 4, 01062 Dresden, Germany. [6]Max Planck Institute for Polymer Research, Ackermannweg 10, 55128 Mainz, Germany. [7]State Key Laboratory of Elemento-Organic Chemistry, College of Chemistry, Nankai University, 300071 Tianjin, China. [8]ZHAW Zurich University of Applied Science, School of Engineering, Institute of Materials and Process Engineering, Technikumstrasse 9, 8401 Winterthur, Switzerland. [9]Physics Department E20, TUM School of Natural Sciences, Technical University of Munich, James-Franck-Str. 1, 85748 Garching, Germany. [10]School of Integrated Circuits and Electronics, MIIT Key Laboratory for Low-Dimensional Quantum Structure and Devices, Beijing Institute of Technology, 100081 Beijing, China. [11]Humboldt-Universität zu Berlin, Department of Physics & Center for the Science of Materials Berlin (CSMB), 12489 Berlin, Germany. [12]These authors contributed equally: Juan Li, Amir Mirzanejad, Wen-Han Dong. ✉e-mail: jtsun@bit.edu.cn; lfm5572@psu.edu; palma@iphy.ac.cn

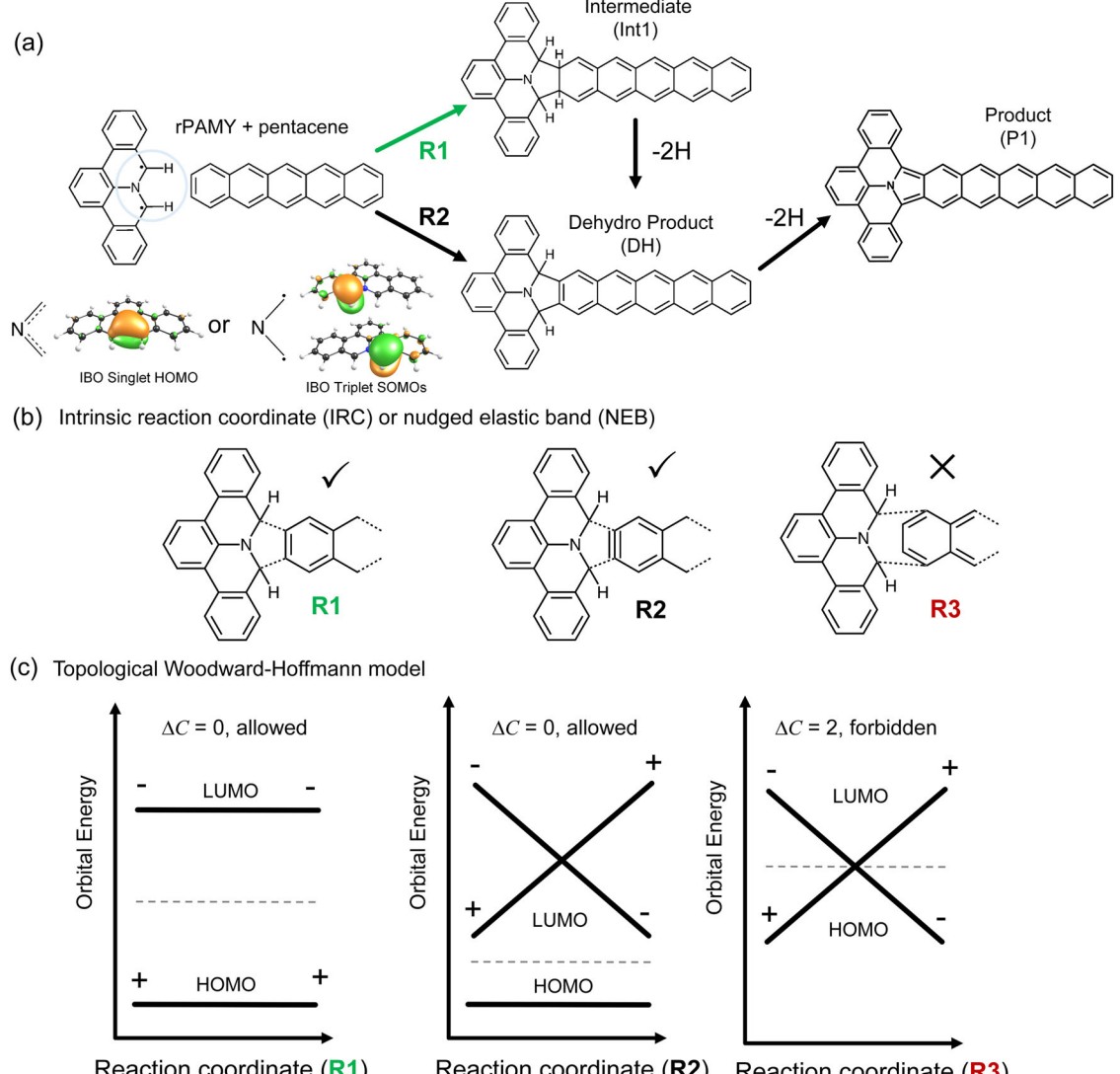

**Fig. 1 | Singlet diradicaloid PAMY cycloadditions and their symmetry-allowed or forbidden pathway classification as a topological obstruction. a** The reaction scheme of diradicaloid PAMY (rPAMY) with pentacene posits few possible reaction pathways, one associated with pentacene de-aromatization intermediate (Int1) and the other without a dehydrogenated intermediate implying pentacene aromaticity-conserving pathways (R2). The rPAMY triplet lies 25 kcal mol⁻¹ above the singlet and hence only the singlet reactive species is considered in this work. The intrinsic bond orbital (IBO) DFT analysis identifies the ground state as a singlet resonant structure of the diradical. **b** Three cycloaddition pathways R1, R2, and R3 between singlet rPAMY and pentacene investigated by means of intrinsic reaction coordinates or nudged elastic band. **c** Topological Woodward-Hoffmann classification. Woodward–Hoffmann (WH) frontier orbital rationalization of allowed or forbidden reactions the where "+" ("–") denotes an even (or odd) MO with respect to a symmetry reference (see Supplementary Figs. 1–2). The dashed line separates occupied (HOMO) from unoccupied orbitals (LUMO). In topological classifications, explicit parameters for the analytical understanding of reaction coordinates, isomeric reaction pathways in a formal mathematical framework for the topological classification of occupied frontier orbital intersections are explored, beyond WH rules alone.

precursor, namely 8*H*-isoquinolino[4,3,2-*de*]phenanthridin-9-ium tetrafluoroborate (DBAP salt, **1**, see Methods and Supplementary Scheme 1), undergoes selective 1,3-dipolar [3 + 2] cycloaddition to electron-deficient dipolarophiles yielding N-containing PAHs[42,43]. Towards the engineering of extended N-containing PAHs on-surface and in the solid-state, a mechanistic, broadly accessible understanding of the chemical reactions and pathways accessible to PAMY and similar cycloadditions is desirable.

In on-surface and solid-state thermochemistry, chemical reaction modeling beyond adiabatic energetic diagrams (e.g. nudged elastic band method for determining transition structure and energy barrier[52]) is rarely explored since the potential energy surfaces of ground states are often assumed to follow the noncrossing rule[53] or void of electronic state hopping. Because of this limitation, it remains often unclear in the literature whether cyclization and dehydrogenation reactions are symmetry-allowed, that is, occur adiabatically or otherwise. Symmetry-forbidden reactions, that is,

reactions in which molecular orbitals cross, are indicative of nonadiabatic dynamics that are known to play a fundamental role in PAH synthesis and cycloadditions[54,55]. Several models are available to attempt to formally define forbidden nonadiabatic reaction pathways from on-surface reaction mechanisms. Woodward and Hoffmann (WH)[56–60] attribute the reactivity of a chemical reaction to the atomic orbital symmetries under adiabatic conditions. The WH rules provide qualitative selection rules for pericyclic[61], that is, concerted cycloaddition reactions[62–66] whereby molecular orbital symmetry and crossings are commonly treated by means of Fukui reaction theory[67], Marcus theory of electron transfer and the surface hopping method[68]. Accurate mechanistic predictions relying on the Born-Oppenheimer[69] approximation, are challenging when dealing with more than one reaction pathway in the presence of nonadiabatic quantum effects such as tunneling and surface hopping. These effects make it difficult to validate qualitative WH rules. Yet the WH approach remains a powerful

well-known concept for reaction engineering. In this regard, quantitative and chemically-intuitive WH visualization tools treating ionic, radical and pericyclic reactions on equal footing would be desirable for the rapid prototyping of PAH reactions, and the study of nonadiabaticity. During the last decades, the classification of orbital crossings[70–72] aided by differential geometry and algebraic topology has emerged as a promising method to study nonadiabaticity in condensed matter physics eigenproblems[73–80]. The topological study of eigenproblems such as band topology, fundamentally differs from various real-space applications of topology, such as the use of quantum chemical topology for bond connectivity[81–83]. Because physical properties are encoded in quantum matter as eigenproblems, topological classification aids in their engineering with an additional design dimension, supported by the interplay between effective ('toy') models, ab initio calculations and experiments[84–88]. Recently, the concept of topology classification for mirror-symmetric reaction pathway models to study reactions by means of topological invariants was introduced[89,90], whereby the reactions with distinct topologically classifiers are adiabatically forbidden (such as the [2 + 2] thermal reaction of two ethylene molecules). Topological classifications could epitomize a turning point to accelerate cycloaddition reaction engineering, especially on-surface[91], where reactions are surface templated and highly symmetric. Particularly, such topological models expand and unify the WH-Fukui approach: They enforce the geometrical symmetry and concertedness of reaction pathways to summarize and formalize chemical notions, simplifying reaction interpreting and rational design. Additionally, they illustrate that reaction coordinates can be mathematically defined to study nonadiabaticity and (non-interacting) orbital intersections from a topological standpoint[86,92].

Here, we study the solid-state and on-surface cycloaddition of a PAMY precursor and pentacene to yield internally N-containing PAH with a tetracenoisoindole core (Fig. 1a, the optimized structures provided in Supplementary Data 1) as characterized by ultra-high vacuum (UHV) scanning tunneling microscopy (STM) and matrix-assisted laser desorption-ionization mass spectrometry (MALDI-MS). By investigating the frontier orbital symmetries of gas-phase reaction pathways by means of intrinsic reaction coordinates which are assumed conclusive for the study of the on-surface and in the solid-state reactions, we describe the [3 + 2] reaction (Fig. 1b) between singlet diradicaloid PAMY (rPAMY) and pentacene and show that its de-aromatization pathway is adiabatic and therefore thermally allowed in the gas phase. We formally classify the WH rules via topological invariants, extending the recently proposed topological classification[89] to a tight-binding *Hückel reaction* model combining first-principles calculations (Fig. 1c). Different from the frontier orbital model, our topological WH model differentiates the allowed, concerted adiabatic pathways, from the nonadiabatic crossings by a $\mathbb{Z}$-classified topological invariant $C(t) = N_+(t) - N_-(t)$, where $N_+(t)$ ($N_-(t)$) is the number of mirror symmetric (antisymmetric) molecular orbitals (MOs) in all occupied MOs, and $\triangle C = C_{react} - C_{prod}$ is the difference of the topological invariants between reactants and products (Fig. 1c). We find that singlet diradicaloid PAMY lateral addition to acenes is endothermic but topologically WH allowed, while central ring addition is exothermic but topological forbidden. Our work introduces a methodological and theoretical approach for the study of cycloaddition selectivity, particularly PAMY reactions which are relevant in development of N-containing PAHs as substrates for N-doped nanographenes, spin chains[50,51,93], metal-free catalysis[94,95] and sensors[96,97].

## Results and Discussion
### Cycloaddition between rPAMY and pentacene: MALDI-MS and STM
Unlike homocoupling solution reactions[49], previous heterocoupling solid-state studies[25] employing DBAP salt have not identified key intermediates which unambiguously evidence cycloaddition reaction pathways. Therefore, we set to investigate on-surface and solid-state intermediates and products between the DBAP and pentacene in Fig. 2 employing mass spectrometry (MS) and STM. Upon heating to 250°C a 1:1 solid-state mixture of pentacene and DBAP salt, a peak of *m/z* = 543.198 is identified in

the matrix-assisted laser desorption/ionization (MALDI) mass spectrum (Fig. 2a), assigned to the heterocoupling product with a hydrogenated tetracenoisoindole core, together with the expected competing homocoupling products[49]. A tentative structure for the product is **DH** (Figs. 1a and 2a) ensuing from a hydrogenated tetracenoisoindole core and partial dehydrogenation of pentacene. Such product could occur following pathway **R1** (Fig. 1a) as a probable adiabatic cycloaddition mechanism. A direct, unknown pathway **R2** could occur but is less plausible. These pathways consider that a hydrogen of the unreactive species of DBAP is removed, to form diradicaloid PAMY (rPAMY, Fig. 1a), which has been previously characterized on surfaces[25,49]. The diradicaloid term[98] is used to address both potential spin states of rPAMY, whereby the singlet and triplet frontier orbitals are depicted in Fig. 1a with intrinsic bond orbital (IBO) analysis. It is worth noting that DFT places the triplet diradical PAMY 25 kcal mol$^{-1}$ higher in energy than the singlet species, which is around 8 kcal mol$^{-1}$ above than the highest energy barrier found in this work, and therefore not considered as the reactive species. When mixing 2.2 equivalents of DDQ in the solid-state reaction to further dehydrogenate the DH intermediate (see Methods), a fully dehydrogenated product, tentative **P1**, bearing a tetracenoisoindole core is identified (Fig. 2b). Note that intrinsic reaction coordinate and broken-symmetry DFT calculations find that the intermediate with two detached hydrogens (DH) is more stable than the final product **P1**, consistent with the MALDI data above (see next section, Fig. 3).

Low temperature (LT) STM measurements on the Ag(100) surface (Fig. 2c-d) were carried out to further investigate surface-confined reaction products. Upon sublimation of pentacene and DBAP onto Ag(100) and annealing treatment at around 400°C, the major product of the reaction is identified as bearing mirror symmetry with the 2,3 position of pentacene as studied in Fig. 3. Metal substrates are known to catalyze dehydrogenation, such that the reaction on Ag(100) is comparable to the solid-state reaction with DDQ and the reaction product is inferred as the expected final product **P1** with a tetracenoisoindole core. Moreover, the three-dimensional structure of **Int1**, with an angle of 125° between the PAMY and pentacene, strongly biases the reaction towards planarization through partial dehydrogenation products **DH** and **P1**. An additional minor product **P2** derived from the pentacene molecules reacting on the 1,2 positions is observed (Supplementary Fig. 13).

### Cycloaddition between rPAMY and pentacene: Intrinsic reaction coordinate calculations
To study isomeric reaction pathways, we focus on the cycloaddition between singlet diradicaloid PAMY (rPAMY), and pentacene by broken-symmetry DFT calculations. Figure 3b depicts the energy profile obtained from intrinsic reaction coordinate (IRC) calculation of the **R1** proposed reaction pathway leading to **P1** via on-surface dehydrogenation as detailed in supporting information. There are at least two additional plausible reaction pathways (**R2-R3**) which could explain the on-surface and solid-state data (Fig. 3). The calculated frontier orbitals of the starting configuration can be defined as symmetric (S) or antisymmetric (AS) with respect to the molecular mirror plane. These symmetries are conserved in the endothermic **R1** [3 + 2] cycloaddition IRC pathway (Fig. 3a, b) and therefore WH allowed. In pathway **R2**, which considers the cycloaddition reaction with dehydrogenated pentacene, the symmetry of the highest occupied molecular orbital is conserved and hence equally WH allowed. Here, attempts to locate a transition structure via quantum mechanical structure optimization were unsuccessful, whereas nudged elastic band identifies the reaction as barrierless and highly exothermic (Fig. 3c, d). Finally, the solid-state reaction could admit an isomer of product **P1** through pathway **R3**, in which orbitals symmetries are changed. Accordingly, the transition structure search fails to locate a concerted [3 + 4] cycloaddition mechanism, despite the **Int2** being exothermic (Fig. 3e, f). In addition to the concerted mechanism being WH forbidden, the pathway towards **Int2** is obstructed through hydrogen migration, see hydrogen in Fig. 3f. Thus, the 7.3 kcal mol$^{-1}$ energy barrier corresponds to the only reaction found, featuring a

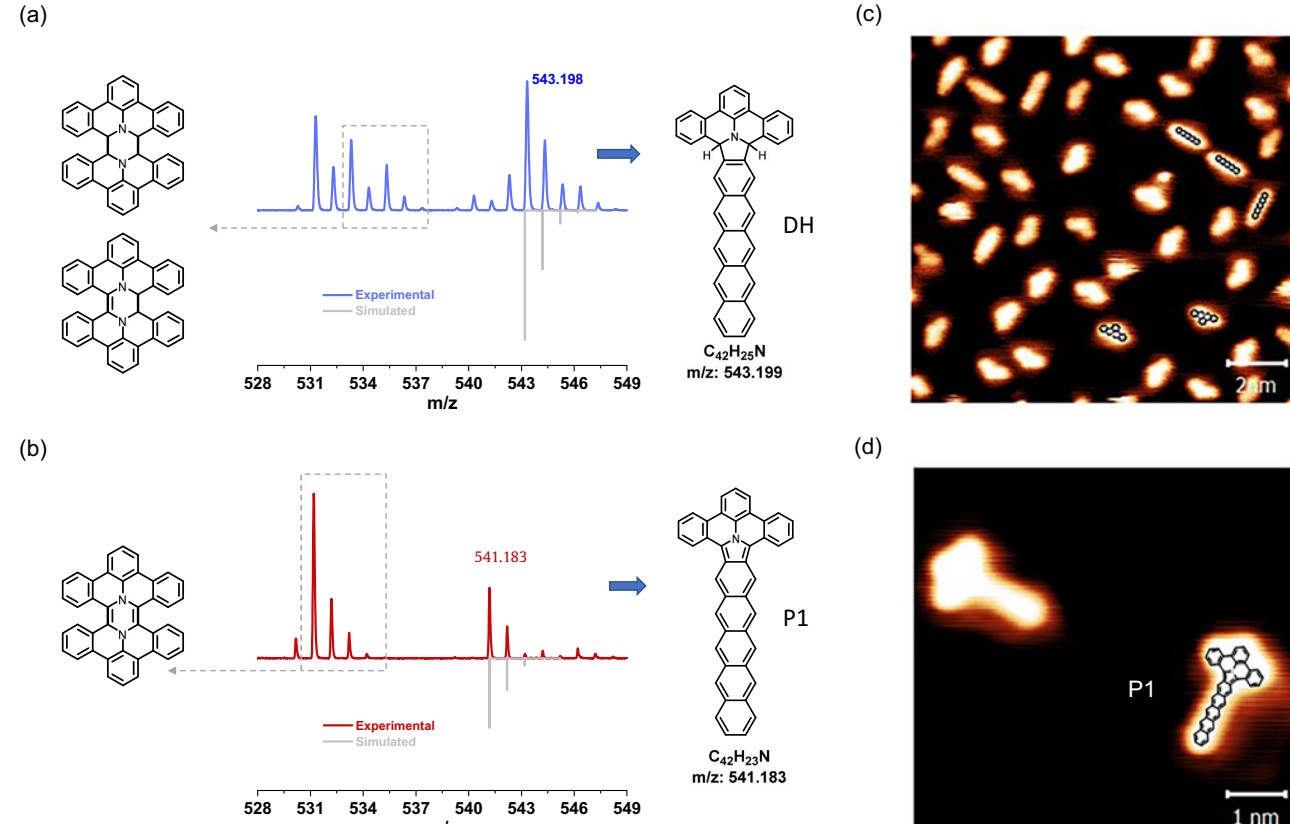

**Fig. 2 | Reaction of rPAMY precursor and pentacene involving de-aromatization of pentacene. a** MALDI-TOF mass spectra of the product from solid-state synthesis at 250°C with DBAP salt and pentacene (1:1). **b** MALDI-TOF mass spectra from solid-state synthesis at 250°C with DBAP salt, pentacene and DDQ (1:1:2.2). The competing products from rPAMY cycloaddition dimerization, diaza-based PAMY dimers, are also shown. **c** STM survey of the DBAP and pentacene reaction after simultaneous evaporation of precursor and pentacene on Ag(100). Scanning parameters: $I_t = 30$ pA, $V_s = 300$ mV. **d** High-resolution STM of the reaction product **P1** on Ag(100) after on-surface synthesis at ~ 400°C. Scanning parameters: $I_t = 30$ pA, $V_s = 300$ mV.

hydrogen migration rather than the concerted 7-member ring addition. This reaction breaking mirror symmetry (cf. next section), is consistent with the step-wise reactivity of pentacene at the 6,13 position[99,100]. Further generalization to acenes in Supplementary Table 1 confirms that the reaction of rPAMY with acenes is a type-I HOMO-controlled [3 + 2] cycloaddition[65] since rPAMY is the electron donor, in line with suggested mechanisms for high-temperature on-surface and in-solution rPAMY dimerization[49] and a recently reported heterocycloaddition, the polymerization of cyano-rPAMY[25] (Supplementary Fig. 10). Classifying forbidden and allowed reaction pathways according to WH rules, offers qualitative criteria for the design of reaction pathways[101], yet a formal, topological classification could qualitative convey the symmetries and obstructions involved in cycloadditions.

**Adiabatic vs. nonadiabatic pathways: Topological classification**
A symmetry forbidden WH pathway can be defined as orbitals crossing, and this crossing or obstruction topologically classified by a non-zero change of topological invariants. In our previous study[89], we introduced the reaction model method to topologically characterize the molecular chemical reactions. This approach employs topological invariants to classify the discontinuities between eigenvector spaces of reaction matrices. In Supplementary Fig. 1, we elaborate on a methodology for the construction of a *Hückel reaction* model. Utilizing this methodology, we construct the mirror-symmetric (i.e. mirror symmetry-enforced) *Hückel reaction* models for the reaction between rPAMY and hydrocarbons, as depicted in Fig. 4. As an extended formal framework for WH rules, topological classification models can handle pericyclic and radical cycloadditions[102–104] and are highly useful for visualizing plausible

reaction pathways with enforced reaction symmetry (mirror plane depicted in Fig. 4a). Compared with classic WH diagrams, this model grants explicit access to the quantitative exploration of concerted cycloadditions which conserve the matrix symmetry by means of tunable bonding strengths. The model could be extended, in principle, to biradical cases[54,55] and complex geometrical transformations of molecules, provided certain symmetries are included.

In a *Hückel reaction*, we construct a general $6 \times 6$ or $8 \times 8$ *Hückel matrix* of the four-atom [3 + 2] or [3 + 4] representation of the reaction between the $p$ orbitals of a rPAMY fragment and butadiene or benzene in Fig. 4a, b or c–e, respectively. Figure 4e shows an $8 \times 8$ *Hückel matrix* for the forbidden [3 + 4] **R3**-like cycloaddition. It shows how the key quantifying reaction coordinate $t$ in the *Hückel reaction* model is inversely related to the distance between the reactants $t = 1/d$, and how crossings are identified through discontinuities in the frontier orbitals and their symmetries. Here, we assume that the valence electrons of the nitrogen are bonded to the carbons with strength $a$, and the onsite energy $p1$ ($p1^*$) differs from $p2$ due to the heteroatom effect. To enforce symmetry and account for both $p$ electrons of nitrogen, we consider a virtual $sp^2$ nitrogen where the lone pair occupies a degenerate pair of $p_z$ and $p_z^*$ orbitals, thereby virtually increasing the order of the cycloaddition from a [3+n] ring to a [4+n] ring. The interactions of the two reactants can be elegantly described by a single variable parameter $t < 0$ with $|t|$ as the reaction coordinate. Forbidden pathways in the model indicated by crossings of molecular orbitals that are Z-classified by the topological invariant $C(t) = N_+(t) - N_-(t)$ (see Supplementary Fig. 2 and previous section). The topological analysis of the frontier orbital evolution in Fig. 4b, d (see details in Supplementary Fig. 2)

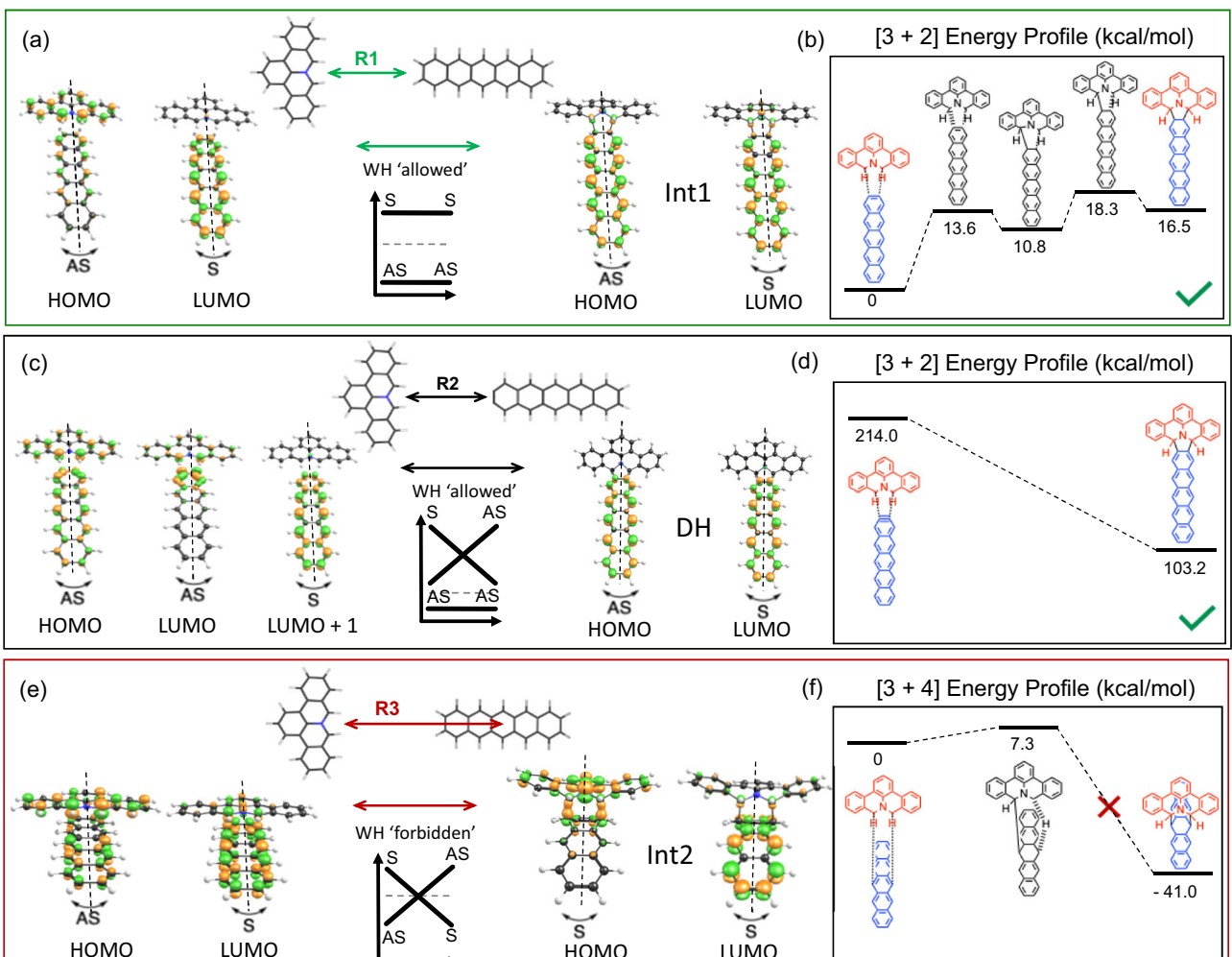

**Fig. 3 | Woodward-Hoffmann exploration of reaction pathways for rPAMY + pentacene cycloadditions which are plausible in the solid-state or on-surface.** **a**-**b** Endothermic PAMY+pentacene **R1** [3 + 2] intrinsic reaction coordinate (IRC) cycloaddition pathway to the intermediate product (Int1) towards fully dehydrogenated **P1**. The frontier orbitals are also depicted. **c**-**d** For comparison the **R2** exothermic [3 + 2] cycloaddition pathway with an aryne moiety in pre-dehydrogenated pentacene is purely concerted and allowed. **e**-**f** The IRC exploration of exothermic PAMY+pentacene **R3** pathway, plausible in the solid-state, shows that a hydrogen migration reaction takes place in lieu of the WH forbidden, obstructed cycloaddition reaction towards **Int2**. The barrier features a hydrogen migration reaction, which strongly breaks mirror-symmetry (cf. Figure 4).

shows that the diradicaloid reaction is allowed with benzene ($\triangle C = 0$), but forbidden with the 1,4-positions of butadiene ($\triangle C = 2$). The Supplementary Figs. 4 and 5 provide details of how to obtain the parameters by fitting the *Hückel*-like parameters with DFT orbitals. The tight-binding *Hückel reaction* **R3**-like pathway is extended with DFT calculations in Fig. 4f, g and the optimized structures are provided in Supplementary Data 1. The DFT reaction pathway is only approximately forbidden, as strict eigenstate crossings seldom occur beyond tight-binding models. In such cases, Green function formalism can be employed to classify topological crossings in highly correlated reaction pathways[105,106]. In summary, cycloaddition WH reaction matrix models help formalize WH rules and vastly extend the scope of the engineering of adiabatically forbidden or allowed radicaloid reaction pathways.

## Conclusions
We studied the on-surface and solid-state reaction of PAMY with pentacene from a Woodward-Hoffmann topological classification point of view, and computationally identified pathway **R1** as most plausible. Pathway **R1** entails a Woodward-Hoffmann topologically allowed, endothermic de-aromatization of pentacene and subsequent dehydrogenation yielding an internally N-containing polycyclic aromatic **P1** on Ag(100) and in the solid-state. In addition, we studied a plausible barrierless pathway **R2**, with pre-dehydrogenated pentacene, opening up avenues for the design of efficient reaction pathways with PAMY. A more reactive pathway **R3**, plausible in the solid-state, is rationalized as Woodward-Hoffmann forbidden from frontier orbital analysis and topologically Woodward-Hoffmann forbidden in a symmetry-enforced tight-binding, *Hückel reaction* model. Topological Woodward-Hoffmann models offer a pedagogic entry to the analytical study of mathematically-defined reaction pathways and radicaloid cycloadditions, and pave the road for the engineering of solid-state or on-surface AB-type cycloaddition polymerization and related nanographenes; building on three levels of interdisciplinarity common in quantum matter design: Topological classification of analytical models, verification or parameterization with quantum chemistry, and experimental realization.

## Methods
### Experimental
The precursor (DBAP molecule) was characterized by NMR-spectroscopy in $d_2$-dichloromethane (SI). The exact molecular weight of DBAP and pentacene additives was detected by high resolution matrix-assisted laser desorption/ionization time of flight (HR-MALDI-TOF) mass spectrometry (MS, see Supplementary Fig. 14).

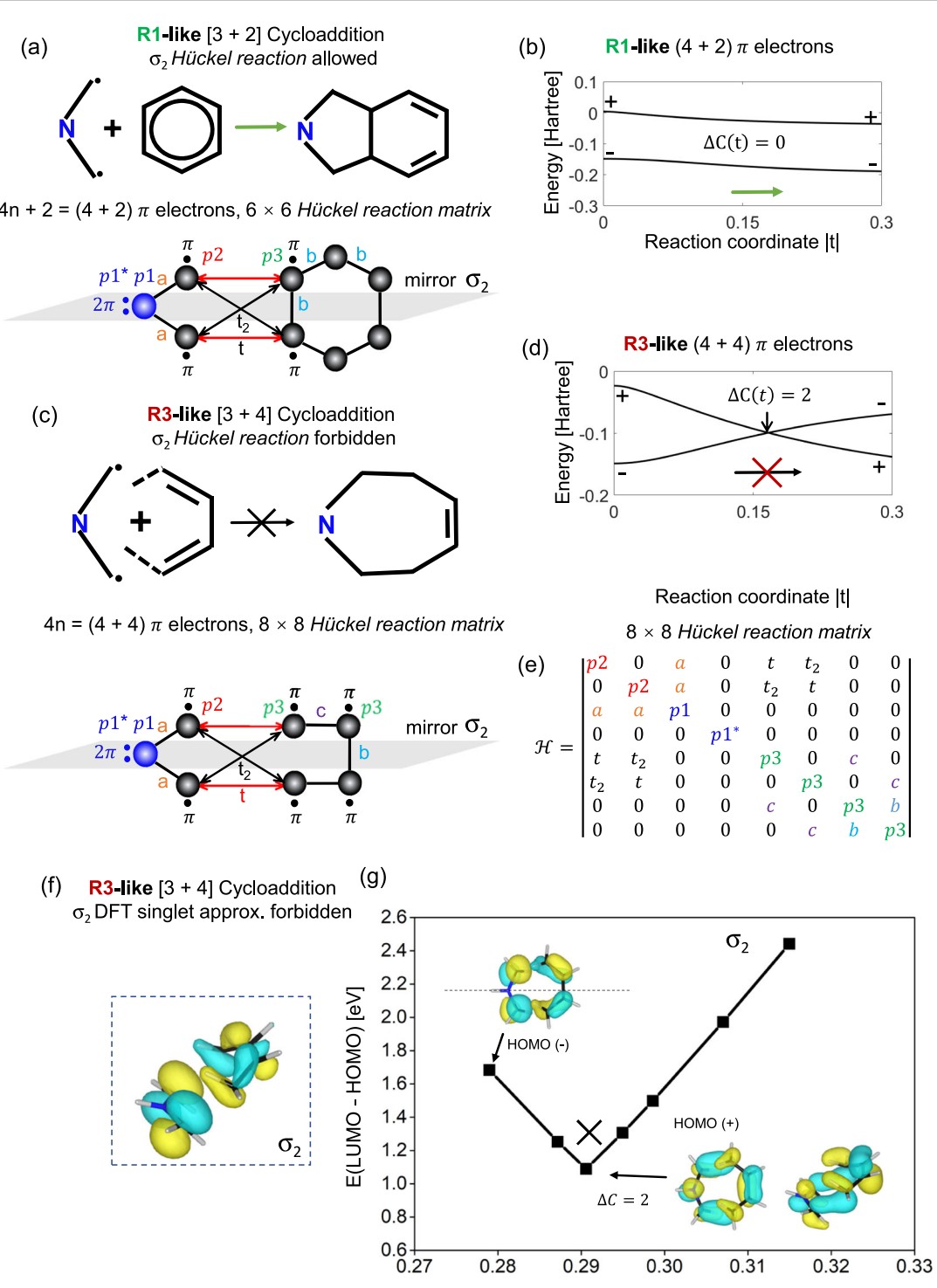

**Fig. 4 | $\sigma_2$–Topological Woodward-Hoffmann classification preserving mirror $C_{2h}$ orbital symmetry for the aziridine+benzene *Hückel reaction* model and DFT parameterization.** **a** The **R1**-like reaction of aziridine diradicaloid as a PAMY model with benzene and the reaction model scheme with $\pi$ electrons as sites. **b** The HOMO and LUMO evolution of (**a**) as t changes, fitting with DFT results: $a = -0.138$, $b = -0.125$, $p1 = -0.216$, $p1^* = 0.065$, $p2 = -0.149$, $p3 = -0.121$, $t_2 = 0.2t$ in units of Hartree. The topological invariant per molecular orbital is $C(t) = N_+(t)-N_-(t)$, see text and Supplementary Fig. 2. **c** The **R3**-like reaction of the diradicaloid with butadiene and the reaction model scheme. **d** The HOMO and LUMO evolution of (**c**) as t changes, fitting with DFT results: $a = -0.138$, $b = -0.088$, $c = -0.141$, $p1 = -0.216$, $p1^* = 0.065$, $p2 = -0.149$, $p3 = -0.127$, $t_2 = 0.2t$ in units of Hartree. The crossing in (**c**) can be classified topologically, Supplementary Figs. 1–2. **e** *Hückel reaction* matrix description of (**c**), with t and $t_2$ as variables. Note that we assume the interaction between bonding orbitals ($p2$) and antibonding orbitals ($p1^*$) is very weak (~0). **f**-**g** DFT study of the reaction modelling in (**c**, **d**) showing an approximate crossing (see text).

For STM characterization, all samples were prepared in a custom-designed UHV chamber with a base pressure below $2.0 \times 10^{-10}$ mbar. The Ag substrate was cleaned by repeated cycles of argon ion sputtering (800 V, 4.5 mA, 15 min) and annealing (flash heating to 450 °C). DBAP and pentacene were deposited on the Ag(100) substrate using the organic molecular beam epitaxy (OMBE) method. The quartz crucibles were held at 300 °C and 180 °C, respectively, with a deposition time of 4 min and a pressure remaining below $5.0 \times 10^{-9}$ mbar. Subsequently, the sample was transferred to the STM. After cooling down to ~ 20 K, constant current STM measurements were performed using a commercial low temperature STM (CreaTec Fischer & Co. GmbH) with a tungsten tip.

## Computational

Automatic quantum mechanical transition search calculations were performed using several initial guesses to determine the transition state structure. This approach could not identify a transition state for highly exothermic R2. Instead, the nudged elastic band method was employed for R2. Broken-symmetry density functional theory (BS-DFT) calculations were performed in gas phase utilizing unrestricted B3LYP functional[107–110] with Grimme's dispersion correction and Becke–Johnson damping [D3(BJ)][111,112]. Full geometry optimization and intrinsic reaction coordinate calculations were initially performed using the minimal basis set[113] and then reoptimized using def2-SVP basis set[114] in ORCA quantum chemistry package[115]. The Intrinsic Bond Orbital (IBO) analysis is a powerful tool for understanding molecular structure and bonding. It localizes molecular orbitals directly from the wavefunction, enabling the interpretation of the electronic structure in chemical systems[116,117]. The intrinsic bond orbital (IBO) analysis was done in PySCF[118] using the wavefunctions obtained in ORCA for singlet and triplet PAMY. The DFT gaps and the reaction energy profile in Supplementary Fig. 10 were obtained using 6-31 G(d) basis set[119] in Gaussian 09 Software[120] by varying inter-molecular distance. The inter-molecular distance was varied in the dimer system by shifting both molecules slightly along the perpendicular direction of the molecular planes to avoid the geometry frustration when they are too close.

The slab calculations involving Ag(100) substrate are performed using Vienna Ab-initio Simulation Package[121] (VASP). The Perdew-Burke-Ernzerhof (PBE) parametrization[122] of generalized gradient approximation (GGA) is adopted for exchange-correlation functional. We considered *van der Waals* (vdW) corrections via Grimme's D3 method[111]. The energy cutoffs of plane-wave basis are set as 400 eV. A single k-point ($\Gamma$) is used for structural optimization and the $k$-mesh of $3 \times 3 \times 1$ was used for the density of states. The convergence criteria for the electronic self-consistent loop and atomic structural optimization are $10^{-6}$ eV for electronic energy and $0.02$ eV Å$^{-1}$ for the atomic force, respectively. The energy profile for detaching one hydrogen atom is calculated in the climb image nudged elastic band (CI-NEB) scheme.

## Data availability

All data supporting the findings of this work are available within this paper and its Supplementary Information. The optimized DFT structures are in Supplementary Data 1. Raw data are available from the corresponding author upon reasonable request.

## Code availability

The central codes used in this paper include Gaussian 09, VASP, ORCA, and PySCF. Detailed information about the license and user guide for these codes are available at https://gaussian.com/glossary/g09/, https://www.vasp.at/, https://www.faccts.de/docs, and https://pyscf.org/.

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

## Acknowledgements
This research was financially supported by the EU Graphene Flagship (Graphene Core 3, 881603), the Chinese Academy of Sciences (nos. QYZDBSSW-SLH038, XDB33000000, XDB33030300), National Natural Science Foundation of China (Grant Nos. 12474178, 11974403, 11974045, 12374172, 62488201, W2532008), the German Research Foundation (DFG) within the Clusters of Excellence "Center for Advancing Electronics Dresden (cfaed)", "Matters of Activity. Image Space Material" EXC 2025 – 390648296, the DFG-SNSF Research Project (EnhanceTopo, No. 429265950), the K. C. Wong Education Foundation of Chinese Academy of Sciences, the National Key Research and Development Program of China (Grant No. 2020YFA0308800), Alexander von Humboldt Foundation and the Guangdong Science Association Young Talent Development Program. We thank Tilo Lübken (Dresden University of Technology) for NMR measurements, Knud Seufert for help with the STM. We gratefully acknowledge Akimistu Narita, Oliver Dumele and Prince Ravat for critical discussions. We acknowledge support by the Open Access Publication Fund of Humboldt-Universität zu Berlin.

## Author contributions
A.M., W.-H.D. and L.M. performed the theoretical analysis. J.L., W.A., J.V.B. and C.-A.P. performed the microscopy measurements and data interpretation. A.M., L.M., W.-H.D., X.-S.D. and J.-T.S. performed and coordinated the calculations. K.L., M.R., X.W., R.B., J.M., A.N., K.M and X.F. performed MS measurements and synthesized the compounds. C.-A.P. designed the research. L.M. and C.-A.P. supervised the project. All authors participated in discussing and editing the manuscript.

## Funding

## Competing interests
The authors declare no competing interests.
