## [Transparent Peer Review file · Communications Chemistry]

This manuscript has been previously reviewed at another Nature Portfolio journal. This document only contains reviewer comments and rebuttal letters for versions considered at Communications Chemistry.

Topological classification of cycloadditions occurring on-surface and in the solid-state

Corresponding Author: Professor Carlos-Andres Palma

Version 0:

Reviewer comments:

Reviewer #1

(Remarks to the Author)

The changes in the manuscript were minor and cosmetic and do not really address our previous comments.

Reviewer #2

(Remarks to the Author)

The requested revision has been successfully carried-out by the authors and manuscript may be considered for publication.

Reviewer #3

(Remarks to the Author)

The authors report synthesis of an N-containing nanographene using a solid state and on-surface synthesis approach. Beside experimental characterization of products by scanning tunneling microscopy and matrix-assisted laser desorption-ionization mass spectrometry, the manuscript extensively discusses reaction selectivity rules for different reaction pathways. This makes the work a highly interdisciplinary contribution involving experimental characterization techniques, quantum chemistry calculations and more pedagogical topological models for the classification of observed reactions. The work has been reviewed beforehand giving rise to detailed suggestions and questions for improving the manuscript at different levels of detail, but also questioning the degree of novelty in view of publication in Nature Communications. Considering the authors' responses to the reviewers and the new version of the manuscript, I feel that all major points have been considered in a convincing way and have been used to further improve the original version of the manuscript. The main points added here are i) details on analysis procedures, ii) clearer statements and citations of earlier work to clarify new contributions by this work and iii) a more accessible classification of the reaction models to make it more accessible for the non-specialist readership.

Overall, I feel that this detailed description of a new type of reaction and the discussion and classification using different reaction models allowing to understand products and reaction pathways is definitively interesting to the community and fulfills criteria for articles to be published in Communications Chemistry. I therefore suggest publication of the new manuscript.

The only point that might be considered by the authors is to somewhat shorten the introduction paragraphs in order to make it more consious by focussing more and aspects that are discussed and concluded in the results/discussion part of the manuscript.

Responses to reviewers

Topological classification of cycloadditions occurring on-surface and in the solid-state

Reviewer #1

The changes in the manuscript were minor and cosmetic and do not really address our previous comments.

Answer: We are grateful to the Reviewer for the time invested in the manuscript. In the first revision, the Reviewer suggested that *“the third (mechanism) would involve a 7-member ring product, which apparently the authors claim is not symmetry allowed but, surprisingly seem to have a lower activation energy than the expected one, which is paradoxical and should be evaluated carefully.”* In response, we clarified that the lower activation energy that we report is not related to a 7-member ring product. We have now written *“(the) energy barrier corresponds to the only reaction found, featuring a hydrogen migration rather than the concerted 7-member ring addition”*. In other words, there was no activation energy found for the symmetry-forbidden mechanism, as suggested by the Reviewer. Further, the Reviewer wrote that *“We disagree that the symmetry orbital considerations for pericyclic reactions is not usually considered, WH rules are and have been a very powerful tool to understand this kind of reactions for nearly 60 years now, however now they complement by other considerations and for good reason which is also present and fundamental for this MS.”* We agreed that WH rules are and have been a very powerful tool to understand these types of reactions in accordance with the abstract and conclusion of our work, and have modified any statements of the contrary accordingly.

Reviewer #2:

The requested revision has been successfully carried-out by the authors and manuscript may be considered for publication.

Answer:

Thank you for your thorough review and valuable feedback on our manuscript. We appreciate your constructive comments in the first revision, whereby it was suggested that *“the manuscript may be more suitable for publication after the above suggested minor revision”*.

The reviewer suggested to: 1. *“Perform several single-point calculations for determining the structure of the transition state and they need to show that they have tried such a manual scan of structure from reactant to product and they also may have to try other methods for locating transition state. (for the exothermic path).”* 2. *“Authors are requested to elaborate more detailed discussion about the construction of Hückel reaction model and reaction matrix”*. Correspondingly, we amended the manuscript during the first revision and now further mention that:

Re. point 1, we have now written in the Methods section that *“Intrinsic reaction coordinate calculations were performed, wherein a semi-manual scan of several calculations for determining the structure of the transition state was performed. This approach could not identify a transition state for highly exothermic R2. Instead, the nudged elastic band method was employed for R2”*.

Re. point 2, thank you for suggesting that an extended discussion would be beneficial around the *Hückel* constructions in Figure 4, Supplementary Figure 1 and Supplementary Figure 3. After careful consideration on how to extend that further discussion might decrease the readability of the manuscript.

Reviewer #3 (Remarks to the Author):

The authors report synthesis of an N-containing nanographene using a solid state and on-surface synthesis approach. Beside experimental characterization of products by scanning tunneling microscopy and matrix-assisted laser desorption-ionization mass spectrometry, the manuscript extensively discusses reaction selectivity rules for different reaction pathways. This makes the work a highly interdisciplinary contribution involving experimental characterization techniques, quantum chemistry

calculations and more pedagogical topological models for the classification of observed reactions.

The work has been reviewed beforehand giving rise to detailed suggestions and questions for improving the manuscript at different levels of detail, but also questioning the degree of novelty in view of publication in Nature Communications. Considering the authors' responses to the reviewers and the new version of the manuscript, I feel that all major points have been considered in a convincing way and have been used to further improve the original version of the manuscript. The main points added here are i) details on analysis procedures, ii) clearer statements and citations of earlier work to clarify new contributions by this work and iii) a more accessible classification of the reaction models to make it more accessible for the non-specialist readership.

Overall, I feel that this detailed description of a new type of reaction and the discussion and classification using different reaction models allowing to understand products and reaction pathways is definitively interesting to the community and fulfills criteria for articles to be published in Communications Chemistry. I therefore suggest publication of the new manuscript.

The only point that might be considered by the authors is to somewhat shorten the introduction paragraphs in order to make it more consious by focussing more and aspects that are discussed and concluded in the results/discussion part of the manuscript.

Answer: Thank you for your thorough review and valuable feedback on our manuscript. We appreciate your constructive comments and support throughout this process.

The introduction section has been shortened and now focuses more closely on the core content of this manuscript.